# Effectiveness of neck-specific exercises with and without internet-based support on dizziness/unsteadiness in chronic whiplash-associated disorders: Secondary analyses of a randomised controlled trial

Anneli Peolsson[1,2]*, Sara Wirqvist[1], Ann-Sofi Kammerlind[3], Gunnel Peterson[1,4]

1 Department of Health, Medicine and Caring Sciences, Unit of Physiotherapy, Linköping University, Linköping, Sweden, 2 Occupational and Environmental Medicine Centre, Department of Health, Medicine and Caring Sciences, Unit of Clinical Medicine, Linköping University, Linköping, Sweden, 3 Futurum, Region Jönköping County, Sweden, 4 Centre for Clinical Research Sörmland, Uppsala University, Eskilstuna, Sweden

* anneli.peolsson@liu.se

## Abstract

### Aim

To investigate the effectiveness in individuals with chronic whiplash-associated disorders (WADs) of neck-specific exercise (NSE) supervised by a physiotherapist twice a week for 12 weeks versus neck-specific exercise with internet support and four physiotherapy visits (NSEIT) regarding dizziness, unsteadiness and balance, and to investigate the differences between WAD grades.

### Method

This is a secondary analysis of a prospective randomised multicentre study (RCT) with concealed allocation (ClinicalTrials.gov Protocol ID: NCT03022812). The outcomes were dizziness measured on the Dizziness Handicap Inventory (DHI); dizziness at rest and during activity and unsteadiness using visual analogue scales; and standing on one leg with eyes closed (SOLEC). Participants (n = 140) were randomised to NSE or NSEIT. Measurements were obtained at baseline, and at three- and 15-month follow-ups by assessor-blinded investigators.

### Results

There were no significant differences between NSEIT and NSE in any of the outcomes (p>0.38). Both NSEIT and NSE improved over time (p<0.02; effect size (ES) = 0.74–1.01) in DHI score and dizziness during activity. There was a significant group-by-time interaction effect in dizziness (at rest: p = 0.035; ES: 0.66; and during activity: p = 0.016; ES: 1.24) between WAD grades. Individuals with WAD grade 3 had dizziness/unsteadiness to a

**Data Availability Statement:** Data cannot be made publicly available for both ethical and legal reasons as public availability would compromise patient privacy and information about their health. Coded data will be received open reasonable request and after ethical permission. Please reach out to: registrator@hmv.liu.se.

**Funding:** The authors of this study report the following sources of funding: The Sweden's Innovation Agency (GP, grant number 2018-02244) The Swedish Research Council (AP, grant number 2018-02476) The Centre for Clinical Research Sörmland at Uppsala University, Sweden (GP, grant number DLL-36751, DLL-854531, DLL-930399, DLL-939813) The Regional Research Council Mid Sweden (GP, grant number 838701) The Medical Research Council of Southeast Sweden (AP, grant number FORSS-939838, FORSS-563491; FORSS-650191) The county council of Östergötland, Sweden, (AP, grant number RÖ-724851, RÖ-602771) Linköping University, Sweden (AP, time for research included in the position as Professor).

**Competing interests:** NO authors have competing interests

greater extent and improved in all outcomes over time (p<0.04) compared to those with WAD grade 2, except for SOLEC.

## Conclusions

There were no significant group differences between NSEIT and NSE. Both groups decreased in terms of self-reported dizziness (DHI, dizziness during activity), with medium to large effect size. Those with WAD grade 3 have dizziness/unsteadiness to a greater extent than those with WAD grade 2. Despite improvements, many participants still reported dizziness at 15-month follow-up, and additional balance training and/or vestibular exercise may be investigated for potential additional effect.

## Introduction

Whiplash-associated disorders (WADs) are a disabling and costly condition after an indirect neck trauma through acceleration/deceleration injury, e.g., after a road traffic accident [1–3]. Approximately half of those injured transit to chronicity [4, 5], with remaining symptoms including neck pain and disability, and perhaps the most disabling symptoms of all (with a major impact on health-related quality of life): unsteadiness and dizziness [6–9].

Individuals with WAD have been shown to have more fatty infiltration in the neck muscles replacing contractile tissue [10–12], especially in the deep neck muscles which contain many proprioceptors [13, 14], and are important for proper neck muscle function and orientation of the head in space. Altered sensorimotor cervical function has been reported in individuals with WAD [7, 15–18]. A mismatch between information from the vestibular and visual systems and information from the neck muscles [6, 19–21] may be a reason for dizziness. According to Treleaven et al. [7], the most common ways to describe dizziness in individuals with chronic WAD were "lightheaded", "unsteady" and "off-balance", namely non-rotatory.

The amount of fatty infiltration has been shown to decrease [22], sensorimotor function to improve [23], pain and neck-specific disability to reduce, and health-related quality of life to improve after a neck-specific exercise programme [24]. Neck-specific exercise in combination with a behavioural approach has been shown to reduce unsteadiness and dizziness more than general physical exercise in chronic WAD grades 2 (self-perceived neck pain and disability plus musculoskeletal finding emanating from the neck in a physical examination) and 3 (as grade 2, plus additional neurological symptoms and findings) [9]. The neck-specific exercise programme was extensive, with visits to a physiotherapy clinic twice a week for three months. More cost-effective treatments are needed in order to be able to use the limited resources for those with the greatest need. Internet-based rehabilitation may be one of the answers, although its effectiveness needs to be investigated.

In a recent randomised controlled multicentre trial with three- and 15-month follow-ups [24], a three-month internet-based neck-specific exercise programme combined with four visits to a physiotherapy clinic (NSEIT) was shown to be non-inferior to the same exercises and information given twice a week (optimum 24 sessions) at a physiotherapy clinic (NSE) in relation to neck pain and neck-specific disability.

The aim of the present study was to investigate the effectiveness of NSE versus NSEIT regarding outcomes of dizziness, unsteadiness and balance. An additional aim was to investigate whether there are differences in the above variables between those with WAD grade 2 and grade 3.

The aims were achieved.

## Material and methods

This was a planned secondary analysis from a prospective randomised controlled multicentre study (RCT) [24, 25]. The data was collected via digital questionnaires and physical tests at baseline and at three- and 15-month follow-ups, and all tests were performed by experienced physiotherapists blinded to allocation. Participants were included after obtaining written and oral informed consent. The study was approved May 18, 2016 (later updates made to increase the geographical area of inclusion) by the regional ethical review board in Linköping, Sweden (Dnr 2016/135–31). The study was registered (ClinicalTrials.gov Protocol ID: NCT03022812, initial release December 20, 2016, protocol contributor and project investigator Peolsson & Peterson) and the study protocol was published prior to data collection [25]. Data was collected in Swedish outpatient care from April 6, 2017 until September 15, 2020. Participants were randomised to two groups: NSEIT or NSE (for the CONSORT flow diagram, see Fig 1). A more detailed description of the study methodology can be found elsewhere [24, 25].

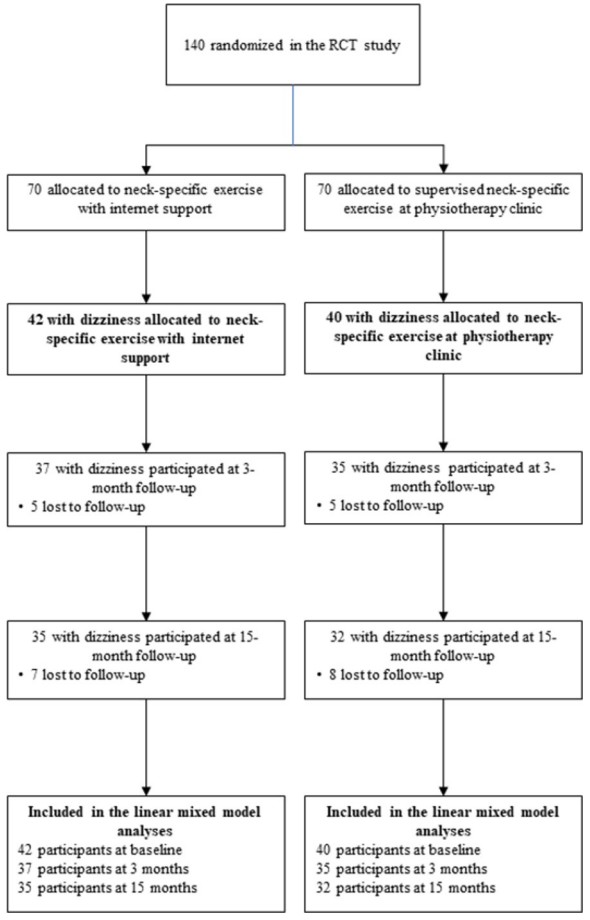

**Fig 1. The CONSORT flow diagram.** (please see ref. 24, for more details about the whole RCT).

## Participants

Participants [24, 25] who reported dizziness and who answered "Yes" to the question "Do you experience dizziness? Yes/No" at baseline were included in these analyses (n = 82 out of 140 in the main study). Inclusion criteria: persistent neck pain ($\geq$ 6 months and $\leq$ 5 years from injury), WAD grades 2 or 3 [1], neck pain in the last week of at least 20 mm on the visual analogue scale (VAS) [26, 27], and neck disability of more than 20% on the Neck Disability Index (NDI) [28]; age 18–63 years; daily access to a computer/tablet/smartphone and the internet; and sufficient time to follow the intervention. Exclusion criteria were any signs of head injury at the time of the whiplash injury; previous fractures or dislocation of the cervical column; serious physical pathology; previous severe neck problems that resulted in sick leave for more than a month in the year before the current whiplash injury; surgery in the cervical column; generalised or more dominant pain elsewhere in the body; other illness/injury that may prevent full participation in the study and/or for which exercises are contraindicated; lack of knowledge of the Swedish language; diagnosed severe mental illness; current alcohol and/or drug abuse; or participation in the earlier NSE study [29].

## Intervention

Participants in the NSEIT group [24, 25] had four visits at the physiotherapy clinic during the 12-week intervention period, with introduction and follow-up for progression of the exercises and to ensure correct performance. The internet programme contained photos (see ref. 24 for photos and explanation of the exercises) and videos of the exercises, and information about relevant musculoskeletal anatomy, whiplash injury mechanisms, neurophysiological and neurobiological pain education, and strategies for dealing with neck pain relapse. Exercises were individually adjusted according to the individual's physical conditions.

In the NSE group [24, 25], participants received the same information and exercises as the NSEIT group, but with supervised training at the physiotherapy clinic twice a week for 12 weeks (optimum 24 sessions).

Pain provocation related to the exercises in NSEIT or NSE was not accepted. At the end of the intervention period, the participants in both groups were encouraged to continue training on their own two or three times a week, and to include NSE in their exercise programme [24, 25].

## Outcomes

**Background characteristics.** Age; sex; current neck pain intensity on VAS [26, 27]; pain duration in months; NDI [28]; and WAD grades [1].

**Outcome data.** Self-perceived problems with dizziness-associated disabilities and handicap on the 25-item Dizziness Handicap Inventory (DHI) instrument [30, 31] (score ranging from 0 to 100, where higher scores indicate more severe problems). The question was only answered by those answering that they experience dizziness ("Yes" from an option of "Yes/no").

Self-perceived dizziness at rest, dizziness during activity, unsteadiness and balance problems, measured on a VAS from 0 to 100 mm, where higher values indicate more problems [26, 27, 32]. Standing on one leg (the dominant leg) with eyes closed (SOLEC), registered in seconds [32]. For safety reasons, the participant stood in a corner with the test leader within reach. The test was interrupted if the participant opened their eyes, reached for support, changed the position of the standing foot, or reached the maximum of 30 seconds. The best out of three measurements was registered [32].

## Statistical methods

Analysis was conducted in cooperation with a statistician according to intention-to-treat principles. Descriptive statistics were used to describe background characteristics and outcome variables. Compliance with exercises was set at attending at least 50% of the recommended exercises (frequency and duration), and was analysed with the Chi-Square test.

Analyses of the continuous outcome variables (DHI, dizziness at rest and during activity, and unsteadiness) were undertaken using linear mixed models (LMMs), with time (baseline, three months, 15 months) and group (NSEIT, NSE) and group by time (baseline, three months, 15 months) as fixed effects. Restricted maximum likelihood estimate was used in the LMMs, allowing all participants with at least one observation to be included in the models, under the assumption of data missing at random. LMMs were also used to analyse the outcome variables in WAD grades 2 and 3, with time (baseline, three months, 15 months) and group (WAD grade 2, WAD grade 3) as fixed effects.

Intention-to-treat analyses used all available data at baseline, three months and 15 months. All participants with at least one observation were included in the models, using restricted maximum likelihood estimate in the LMMs with Bonferroni correction on all multiple pairwise contrasts between the three timepoints.

Differences between groups in SOLEC were analysed with the Mann-Whitney U test. The McNemar-Bowker test was used to determine differences from pre-post for the SOLEC test (within-group analyses), and was used because 14 participants reached 29–30 seconds at baseline and had no margin for improvements. Data was divided into four levels: 0–10 s, 11–20 s, 21–28 s and 29–30 s. The reason for a fourth level (29–30 s) was to obtain information about participants reaching–or close to–normal values for the SOLEC test. The test compares the number of participants who changed their balance in one direction (increased balance) or in the opposite direction (decreased balance).

Missing pattern analyses were conducted both within and between the NSEIT and NSE groups on each outcome, comparing completers and non-completers (a total of eight participants in each group, with missing data at three- and/or 15-month follow-up). For baseline variables and outcome variables at baseline, at three months, and changes from baseline to three months, independent tests were used. At 15-month follow-up, only one participant's data was missing, thus no missing pattern analyses could be conducted. A p-value $< 0.05$ was regarded as significant. Sample size and power calculations were carried out using Power Analysis and Sample Size (PASS, 13.0.8) software based on the primary outcome Neck Disability Index (NDI) in the RCT before data collection started [24, 25]. To detect a between-group noninferiority margin of 7 (percentage units) with 1-sided $\alpha$ = .025 and $\beta$ = .8, a total of 47 participants were needed in each group. To account for attrition, 70 participants were included in each group. The standardized between-group effect, that is, effect size at 3 and 15 months, was calculated as the mean differences between NSEIT and NSE divided by the pooled standard deviation (the weighted average of each group's standard deviation). The standardized margin was calculated as the predefined noninferiority margin divided by the pooled standard deviation. NSEIT is considered noninferior to NSE in NDI if the lower limit (the 1-sided 95% CI of the effect size) does not exceed the standardized margin. Power calculation on secondary outcomes were not performed.

Statistical Package for the Social Sciences (SPSS) version 28.0.0 was used for the analyses.

Cohen's d effect size was used and interpreted as d > 0.2 = a small effect, d > 0.5 = a medium effect and d > 0.8 = a large effect [33].

## Results

Of the total 140 participants in the RCT, 82 (59%) rated self-perceived dizziness, of which 42 were in the NSEIT group and 40 were in the NSE group. There were no significant differences in baseline variables between the groups ($p > 0.29$) (Table 1), but there were significant differences between participants with (n = 82) and without dizziness (n = 58) in NDI and WAD grade. Participants reporting dizziness at baseline had higher NDI, and more of them had WAD grade 3 compared to those without dizziness at baseline ($p < 0.03$) (Table 2). At three- and 15-month follow-ups, 81% and 70% respectively of those with dizziness at baseline had persistent problems of dizziness/unsteadiness. There was no difference in exercise compliance between the groups (NSEIT: n = 29, 81%; NSE: n = 33, 94%, $p = 0.15$) during the 12-week intervention period. The missing pattern analyses regarding completers and non-completers (eight participants in each intervention group, who dropped out at either three- or 15-month follow-up or both) revealed no significant differences in any outcomes between the NSEIT and NSE groups ($p > 0.10$).

### Between group (NSEIT, NSE) differences in DHI score, dizziness at rest, dizziness during activity and unsteadiness

There was no significant main effect of group (NSEIT/NSE) on DHI score ($p = 0.52$), or on self-perceived dizziness (at rest: $p = 0.938$; during activity: $p = 0.384$; unsteadiness: $p = 0.399$). There was no significant group-by-time interaction effect on DHI score ($p = 563$) or on self-perceived dizziness (at rest: $p = 0.938$; during activity: $p = 0.787$; unsteadiness: $p = 0.686$) (Tables 3 and 4).

There were no significant differences between the groups in SOLEC at three- or 15-month follow-ups ($p > 0.34$).

### Between group (WAD grades 2 and 3) differences in DHI score, dizziness at rest, dizziness during activity, and unsteadiness

There was no significant main effect of group (WAD grade 2/WAD grade 3) in DHI score ($p = 0.369$) or in self-perceived dizziness (at rest: $p = 0.464$; during motion/activity: $p = 0.539$; unsteadiness: $p = 0.709$).

**Table 1. Baseline descriptive characteristics of trial participants with dizziness, by treatment group.**

|  | NSEIT (n = 42) | NSE (n = 40) | p |
|---|---|---|---|
| **Age** (years), mean (SD) | 40.6 (12.3) | 39.7 (11.9) | 0.749 |
| **Months since injury**, mean (SD) | 28.8 (20.0) | 28.4 (16.4) | 0.92 |
| **WAD grade**, n (%) |  |  |  |
| grade 2 | 26 (62%) | 20 (50%) | 0.374 |
| grade 3 | 16 (38%) | 20 (50%) | 0.374 |
| **Female**, n (%) | 35 (83%) | 40 (80%) | 0.817 |
| **Male**, n (%) | 7 (17%) | 8 (20%) | 0.817 |
| **NDI**, mean (SD) | 41.2 (11.4) | 38.5 (12.0) | 0.292 |
| **Neck pain now**, mean (SD) | 35.6 (21.2) | 36.2 (22.9) | 0.905 |

NSEIT: neck-specific exercise with internet support

NSE: neck-specific exercise at physiotherapy clinic

WAD: whiplash-associated disorder, grade 2: neck pain and musculoskeletal findings during a physical examination

Grade 3: as grade 2, but with additional neurological findings

NDI: Neck Disability Index (0–100% scale)

Neck pain now: visual analogue scale (VAS; 0–100)

**Table 2. Baseline descriptive characteristics of trial participants (n = 140) with and without dizziness.**

| | Dizziness (n = 82) | No dizziness (n = 58) | p |
|---|---|---|---|
| **Age** (years), mean (SD) | 40.1 (12.0) | 40.8 (10.6) | 0.712 |
| **Months since injury**, mean (SD) | 28.6 (18.2) | 27.9 (16.4) | 0.806 |
| **WAD grade**, n (%) | | | |
| grade 2 | 46 (56%) | 43 (74%) | **0.022** |
| grade 3 | 36 (44%) | 15 (26%) | **0.022** |
| **Female**, n (%) | 67 (82%) | 43 (74%) | 0.302 |
| **Male**, n (%) | 15 (18%) | 15 (26%) | 0.302 |
| **NDI**, mean (SD) | 39.9 (11.7) | 35.3 (10.9) | **0.020** |
| **Neck pain now**, mean (SD) | 35.9 (21.9) | 38.9 (22.2) | 0.439 |

NSEIT: neck-specific exercise with internet support

NSE: neck-specific exercise at physiotherapy clinic

WAD: whiplash-associated disorder, grade 2: neck pain and musculoskeletal findings during a physical examination

Grade 3: as grade 2, but with additional neurological findings

NDI: Neck Disability Index (0–100% scale)

Neck pain now: visual analogue scale (VAS; 0–100)

There was a significant group-by-time interaction effect in self-perceived dizziness (at rest: p = 0.035; ES: 0.66, and during activity: p = 0.016; ES: 1.24). Patients with WAD grade 3 improved in dizziness at rest between baseline and three-month follow-up (p = 0.009), and improvement was sustained to 15-month follow-up (p = 0.047; ES: 0.69). Patients with WAD grade 2 improved between three- and 15-month follow-ups (p = 0.023; ES: 0.77). Patients with WAD grade 3 also improved significantly more in dizziness during activity at three-month follow-up compared to patients with WAD grade 2 (p = 0.036; ES: 0.64) (Tables 5 and 6). There was no significant group-by-time interaction effect on DHI score (p = 0.293) or in self-perceived unsteadiness (p = 0.101). There were no significant differences between the groups in SOLEC at three- or 15-month follow-ups (p > 0.52).

## Within-group effects

Within-group results from the linear mixed model analyses are shown in Tables 3 and 5. Within-group effects are shown in Tables 4 and 5. For the SOLEC test, there were no changes over time (Tables 7 and 8).

There were significant within-group effects in the NSEIT group, with improvements in DHI score and dizziness during activity (p < 0.01). In the NSE group, significant within-group effects were seen in DHI score, dizziness during activity and unsteadiness (p < 0.02). In the NSEIT group, improvements in the outcomes were seen from baseline to both three- and 15-month follow-ups to a greater extent compared to the NSE group, except for unsteadiness which only improved in the NSE group.

There were significant within-group effects for participants with WAD grade 2 in dizziness during activity (p < 0.05). Participants with WAD grade 3 showed significant within-group effects on DHI score, dizziness during activity and unsteadiness (p < 0.04). When NSEIT and NSE were combined into one group, there was a significant improvement in dizziness at rest (p<0.01). However, when analysed separately by group, the results were non-significant (NSEIT p<0.086, NSE p<0.066).

There were no significant within-groups results in the intervention or WAD groups in SOLEC at three- and 15-month follow-ups (p > 0.19).

**Table 3. Linear mixed model results, fixed effects for time points, group, time points x group and univariate effects in the neck-specific exercise with internet support (NSEIT) group and the neck-specific exercise at a physiotherapy clinic (NSE) group.**

| Outcome | Test | F-statistic | p-value | ES |
|---|---|---|---|---|
| **DHI** | Time points | F(2, 69.6) = 8.88 | **< .001** | 1.01 |
| | Group | F(1, 78.0) = 0.42 | .520 | |
| | Time points × group | F(2, 69.6) = 0.58 | .561 | |
| | Time points within NSEIT | F(2, 69.4) = 4.93 | **< .01** | 0.75 |
| | Time points within NSE | F(2, 69.3) = 4.52 | **< .014** | 0.72 |
| **Dizziness at rest** | Time points | F(2, 63.7) = 5.04 | **< .001** | 0.80 |
| | Group | F(1, 76.3) = 0.01 | .938 | |
| | Time points × group | F(2, 63.7) = 0.35 | .708 | |
| | Time points within NSEIT | F(2, 61.7) = 2.55 | .086 | |
| | Time points within NSE | F(2, 63.2) = 2.83 | .066 | |
| **Dizziness during motion/activity** | Time points | F(2, 68.6) = 13.37 | **< .001** | 1.25 |
| | Group | F(1, 76.6) = 0.77 | .384 | |
| | Time points × group | F(2, 68.6) = 0.24 | .787 | |
| | Time points within NSEIT | F(2, 66.5) = 7.02 | **.002** | 0.92 |
| | Time points within NSE | F(2, 68.7) = 6.65 | **.002** | 0.88 |
| **Unsteadiness** | Time points | F(2, 67.9) = 6.78 | **.002** | 0.89 |
| | Group | F(1, 75.9) = 0.721 | .399 | |
| | Time points × group | F(2, 67.9) = 0.38 | .686 | |
| | Time points within NSEIT | F(2, 66.9) = 2.20 | .118 | |
| | Time points within NSE | F(2, 68.5) = 4.75 | **.012** | 0.74 |

NSEIT: neck-specific exercise with internet support

NSE: neck-specific exercise at physiotherapy clinic

DHI: Dizziness and Handicap Inventory, score rating 0–100; a higher score means more severe problems

Dizziness at rest, dizziness during activity and unsteadiness: visual analogue scale (VAS 0–100); a higher value means more dizziness

ES: effect size, Cohen's d

## Discussion

This is the first study investigating the effectiveness of a neck-specific exercise programme on dizziness/unsteadiness/balance for individuals with chronic WAD grades 2 and 3 using internet support, and comparing the extent of dizziness/unsteadiness and balance between WAD grades 2 and 3. To summarise the findings from the present study, there were no between-group differences regarding intervention groups, and only DHI score and self-perceived dizziness during activity improved over time after NSEIT or NSE with medium to large effect. When the NSEIT and NSE groups were combined into one group, participants also improved in unsteadiness (large effect), but not when analysed as separate groups, possibly due to a lack of power. Fifty-nine percent of participants included in the RCT had dizziness or unsteadiness at baseline. Those with WAD grade 3 had a higher degree of dizziness/unsteadiness compared with grade 2. Despite a reduced frequency of dizziness/unsteadiness, a majority (70%) still had remaining dizziness at the 15-month follow-up. There was a significant group-by-time interaction effect in self-perceived dizziness at rest and during activity between WAD grades. WAD grade 3 improved in all outcomes over time, except for SOLEC, while WAD grade 2 improved in dizziness at rest and during activity.

The lack of between-group differences confirms the main outcome findings previously reported by Peterson et al. [24], showing NSEIT to be non-inferior to NSE in pain and disability, and both groups to be improved. The results of the variables reported in the present study

**Table 4. Outcomes in DHI score, dizziness and unsteadiness at baseline and at three- and 15-month follow-ups for each intervention group, and within-group effects.**

| Outcome | Group | Time points, mean (95% CI); p-value | | | Within-group change, mean (95% CI); p-value | |
|---|---|---|---|---|---|---|
| | | Baseline | Three months | 15 months | Three months–baseline | 15 months–baseline |
| **DHI** | NSEIT | 35.1 (29.4 to 40.7) | 25.7 (19.3 to 32.0) | 24.8 (17.9 to 31.8) | -9.4 (-16.0 to -2.8); **p = 0.006** | -10.2 (-17.2 to -3.2); **p = 0.005** |
| | NSE | 31.8 (26.0 to 37.6) | 24.8 (17.9 to 31.7) | 20.8 (13.6 to 28.0) | -6.1 (-12.8 to 0.7); p = 0.08 | -10.9 (-18.2 to -3.6); **p = 0.004** |
| **Dizziness at rest** | NSEIT | 17.9 (12.1 to 23.6) | 12.5 (11.3 to 23.3) | 10.5 (5.6 to 15.4) | -5.4 (-10.6 to -0.1); **p = 0.045** | -7.4 (-14.1 to -0.7); **p = 0.03** |
| | NSE | 17.3 (11.4 to 23.2) | 13.5 (8.9 to 18.5) | 9.3 (4.1 to 14.6) | -3.8 (-9.4 to 1.8); p = 0.183 | -8.0 (-15.0 to -1.0); **p = 0.026** |
| **Dizziness during activity** | NSEIT | 37.0 (29.9 to 44.1) | 24.8 (18.8 to 30.9) | 23.9 (16.7 to 31.2) | -12.2 (-20.5 to -3.9); **p = 0.002** | -13.1 (-21.1 to -5.0); **p = 0.002** |
| | NSE | 33.9 (26.6 to 41.3) | 22.6 (16.0 to 29.2) | 18.7 (10.9 to 26.4) | -11.3 (-20.3 to -2.4); **p = 0.008** | -15.2 (-23.8 to -6.6); **p<0.001** |
| **Unsteadiness** | NSEIT | 31.1 (24.4 to 37.8) | 23.9 (17.1 too 30.7) | 25.7 (17.6 to 33.9) | -7.2 (-15.7 to 1.2); p = 0.119 | -5.4 (-13.5 to 2.7); p = 0.189 |
| | NSE | 30.3 (23.3 to 37.3) | 18.9 (11.5 to 26.2) | 20.6 (22.9 to 29.3) | -11.4 (-20.6 to -2.2); **p = 0.01** | -9.7 (-18.4 to -1.0); **p = 0.029** |
| **SOLEC** | NSEIT | 10.5 (6.0–29.0) | 10.5 (6.0–25.7) | 14.0 (7.0–24.5) | 0 (-3.0–2.7); p = 0.993 | 0 (-2.5–4.5); p = 0.321 |
| | NSE | 10.0 (5.25–19.7) | 18.5 (6.25–30.0) | 14.0 (6.0–30.0) | 2.4 (-0.7–13.7); **p = 0.012** | 2.0 (-1.5–9.5); p = 0.056 |

Data in DHI, dizziness and unsteadiness, mean (95% confidence interval); data in SOLEC, median (interquartile range)

NSEIT: neck-specific exercise with internet support

NSE: neck-specific exercise at physiotherapy clinic

DHI: Dizziness and Handicap Inventory, score rating 0–100; a higher score means more severe problems

Dizziness at rest, dizziness during activity and unsteadiness: visual analogue scale (VAS 0–100); a higher value means more dizziness

Within-group change: a negative value means that DHI dizziness, unsteadiness and balance decreased, and vice versa

can be compared with the study by Treleaven et al. [9] investigating the effectiveness of neck-specific exercise with or without a behavioural approach against general physical activity in chronic WAD grades 2 and 3, using nearly the same neck-specific exercise programme, but without internet support [9, 16, 34–37]. In the study by Treleaven et al. [9], there were differences between the groups (neck-specific exercise with/without a behavioural approach) for the University of California, Los Angeles Dizziness Questionnaire (UCLA-DQ) and dizziness during activity on a VAS. Only the neck-specific exercise group with an additional behaviour programme improved over time (UCLA-DQ, dizziness during activity and at rest), although there was a tendency for improvement in the neck-specific exercise group [9]. In the present study, we added two additional ventral exercises to increase ventral neck muscle endurance. Ventral neck muscle endurance was significantly improved (not seen in the previous study), and may have been helpful in reducing neck-related dizziness. Treleaven et al. [37] reported that dizziness following neck-specific exercise for chronic WAD post-follow-up was associated with a lack of improvement in NDI and neck muscle endurance in flexion, which supports the hypotheses of the advantage of increased neck muscle endurance. In the present study, we also included behavioural components in the information. This may have reassured the patients and decreased their worries, which have been shown to be related to pain and disability [38, 39], and–in the former study by Treleaven et al. [9]–to reduce dizziness. However, the results of dizziness at rest/during activity in millimetres on a VAS in the present study were quite similar to the results reported by Treleaven et al. [9].

Minimal important clinical change for chronic WAD is lacking for the variables used in the present study. For DHI score, a change of 18p has been suggested for individuals with dizziness [30], but this has been questioned by others [31].

In the present study, the effect sizes regarding within-group changes over time were medium to large, and the intervention outcome may therefore be of clinical importance.

In the present cohort, WAD during activity was rated as a problem more than being at rest, which is not surprising since activities such as grocery shopping may involve neck activities

**Table 5. Linear mixed model results, fixed effects for time points, group, time points x group and univariate effects in the two groups WAD grades 2 and 3.**

| Outcome | Test | F-statistic | p-value | ES |
|---|---|---|---|---|
| **DHI** | Time points | F(2, 69.4) = 9.56 | < .001 | 1.05 |
| | Group | F(1, 77.9) = 0.82 | .369 | |
| | Time points × group | F(2, 69.4) = 1.25 | .293 | |
| | Time points within WAD grade 2 | F(2, 69.8) = 2.2 | .115 | |
| | Time points within WAD grade 3 | F(2, 68.4) = 8.33 | < .001 | 0.97 |
| **Dizziness at rest** | Time points | F(2, 63.8) = 4.71 | **.012** | 0.77 |
| | Group | F(1, 76.1) = 0.54 | .464 | |
| | Time points × group | F(2, 63.8) = 3.5 | **.0035** | 0.66 |
| | Time points within WAD grade 2 | F(2, 63.9) = 4.69 | **.013** | 0.77 |
| | Time points within WAD grade 3 | F(2, 62.1) = 3.68 | **.031** | 0.69 |
| **Dizziness during motion/activity** | Time points | F(2, 68.2) = 13.16 | < .001 | 1.25 |
| | Group | F(1, 76.0) = 0.38 | .539 | |
| | Time points × group | F(2, 68.2) = 4.41 | **.016** | 1.24 |
| | Time points within WAD grade 2 | F(2, 68.9) = 6.3 | **.003** | 0.66 |
| | Time points within WAD grade 3 | F(2, 66.6) = 11.16 | < .001 | 0.64 |
| **Unsteadiness** | Time points | F(2, 67.8) = 6.82 | **.002** | 0.90 |
| | Group | F(1, 75.8) = 0.14 | .709 | |
| | Time points × group | F(2, 67.8) = 2.37 | .101 | |
| | Time points within WAD grade 2 | F(2, 69.4) = 1.18 | .312 | |
| | Time points within WAD grade 3 | F(2, 68.4) = 7.81 | < .001 | 0.96 |

NSEIT: neck-specific exercise with internet support

NSE: neck-specific exercise at physiotherapy clinic

DHI: Dizziness and Handicap Inventory, score rating 0–100; a higher score means more severe problems

Dizziness at rest, dizziness during activity and unsteadiness: visual analogue scale (VAS 0–100); a higher value means more dizziness

WAD 2: whiplash-associated disorder, grade 2; neck pain and musculoskeletal findings during a physical examination

WAD 3: whiplash-associated disorder, grade 3; as grade 2, but with additional neurological findings

ES: effect size, Cohen's d

that are problematic for those with chronic WAD. The dizziness during activity may be more strongly related to neck disability and neck movements than dizziness at rest, and is therefore a factor that can be improved with neck-specific exercise. Dizziness during neck activities also improved in the present study.

Neck-specific exercise aims to improve sensorimotor control and alter proprioception, in order to increase neck muscle endurance and postural control to reduce pain and disability, addressing factors that may be of importance for reducing both pain and dizziness [7, 13, 14, 40–43]. In the present study, all individuals had neck pain and disability verified as emanating from the neck, most likely with cervicogenic dizziness, although other causes cannot be excluded. For example, as acceleration of the neck is accompanied by acceleration of the head, the trauma might have caused not only neck injury but also vestibular system dysfunction [44]. Concussive injuries of the brain or whiplash injuries of the neck may also trigger the development of persistent postural perceptual dizziness [45]. To further reduce dizziness and balance/unsteadiness problems, it may be an advantage to combine neck-specific exercise with balance training and/or a vertigo exercise programme, with the primary aim of reducing dizziness [46, 47] for the dizzy/unsteady group. Kamper et al. [48] showed that most improvement is achieved in the initial acute stage of WAD, and that chronic WADs are considered difficult

**Table 6. Outcomes in DHI score, dizziness and unsteadiness at baseline and at three- and 15-month follow-ups for each WAD group, and within-group effects.**

| Outcome | WAD grade | Time points, mean (95% CI); p-value | | | Within-group change, mean (95% CI); p-value | |
|---|---|---|---|---|---|---|
| | | Baseline | Three months | 15 months | Three months–baseline | 15 months–baseline |
| **DHI** | WAD 2 (n = 46) | 29.9 (24.6to 35.3) | 25.4 (19.1 to 31.7) | 22.6 (15.6 to 29.5) | -4.5(-11.0 to 1.9); p = 0.17 | -7.4 (-14.3 to -0.4); **p = 0.038** |
| | WAD 3 (n = 36) | 37.9 (31.9 to 43.9) | 26.3 (19.7 to 32.9) | 23.7 (16.5 to 30.9) | -11.6 (-18.4 to -4.8); **p<0.001** | -14.2 (-21.4 to -7.0); **p<0.001** |
| **Dizziness at rest** | WAD 2 (n = 46) | 17.7 (12.2 to 23.3) | 16.0 (11.4 to 20.6) | 10.0 (5.1 to 14.9) | -1.7 (-7.0 to 3.5); p = 0.551 | -7.7 (-14.3 to 1.1); **p = 0.023** |
| | WAD 3 (n = 36) | 17.3 (11.1 to 23.6) | 9.8 (5.0 to 14.7) | 10.0 (4.9 to 15.3) | -7.5 (-13.0 to -1.9); **p = 0.009** | -7.3 (-14.4 to -0.1); **p = 0.047** |
| **Dizziness during activity** | WAD 2 (n = 46) | 35.2 (28.3 to 42.1) | 28.3 (22.4 to 34.2) | 21.1 (13.8 to 28.4) | -6.9 (-13.6 to 0.2); **p = 0.044** | -14.1 (-22.1 to -6.1); **p<0.001** |
| | WAD 3 (n = 36) | 35.8 (28.1 to 43.5) | 19.1 (12.9 to 25.3) | 22.3 (14.6 to 30.0) | -16.7 (-23.7 to -9.6); **p<0.001** | -13.5 (-22.1 to -4.9); **p = 0.003** |
| **Unsteadiness** | WAD 2 (n = 46) | 29.6 (23.1 to 36.0) | 25.3 (18.5 to 32.1) | 23.5 (15.4 to 31.6) | -4.3 (-11.2 to 2.7); p = 0.672 | -6.0 (-14.2 to 2.01); p = 0.144 |
| | WAD 3 (n = 36) | 32.1 (24.8 to 39.4) | 17.9 (10.8 to 25.0) | 23.5 (14.8 to 32.1) | -14.2 (-21.3 to -7.0); **p<0.001** | -8.6 (-17.3 to 0.04); p = 0.051 |
| **SOLEC** | WAD 2 (n = 46) | 10.0 (5.75–22.75) | 10.0 (6.0–29.0) | 14.0 (7.0–26.5) | 0 (-3.0–7.5); p = 0.136 | 1 (-1.5–6.0); p = 0.073 |
| | WAD 3 (n = 36) | 11.5 (6.0–27.5) | 18.0 (7.8–30.0) | 14.0 (6.0–26.0) | 1.0 (-3.0–4.0); p = 0.201 | 1.0 (-2.5–6.0); p = 0.277 |

Data in DHI, dizziness and unsteadiness, mean (95% confidence interval); data in SOLEC, median (interquartile range)

WAD: whiplash-associated disorder

WAD 2: whiplash-associated disorder, grade 2; neck pain and musculoskeletal findings during a physical examination

WAD 3: whiplash-associated disorder, grade 3; as grade 2, but with additional neurological findings

DHI: Dizziness and Handicap Inventory, score rating 0–100; a higher score means more severe problems

Dizziness at rest, dizziness during motion/activity and unsteadiness: visual analogue scale (VAS 0–100); a higher value means more dizziness

Within-group change: a negative value means that DHI dizziness, unsteadiness and balance decreased, and vice versa

**Table 7. Frequency distribution of SOLEC at baseline and follow-ups in the NSEIT and NSE groups.**

| Group | | | Changes between baseline and three-month follow-up | | | Changes between baseline and 15-month follow-up | | |
|---|---|---|---|---|---|---|---|---|
| | | | N[a] | Deteriorated[b] | No changes[c] | Improved[d] | Deteriorated[b] | No changes[c] | Improved[d] |
| **NSEIT** | **SOLEC, seconds at baseline** | 0–10 s | 21 | 0 | 15 | 6 | 0 | 11 | 7 |
| | | 11–20 s | 7 | 4 | 2 | 1 | 2 | 3 | 1 |
| | | 21–28 s | 2 | 0 | 2 | 0 | 0 | 1 | 0 |
| | | 29–30 s | 10 | 3 | 7 | 1 | 3 | 5 | 0 |
| **NSE** | **SOLEC, seconds at baseline** | 0–10 s | 18 | 0 | 10 | 8 | 0 | 7 | 8 |
| | | 11–20 s | 7 | 3 | 2 | 2 | 4 | 3 | 1 |
| | | 21–28 s | 3 | 1 | 0 | 2 | 1 | 0 | 0 |
| | | 29–30 s | 4 | 1 | 3 | 1 | 0 | 3 | 0 |

SOLEC at baseline is presented in four intervals: number of individuals reaching 0–10 seconds, 11–20 seconds, 21–28 seconds and 29–30 seconds

[a]N: number of participants in each interval at baseline

[b]Deteriorated: number of participants with deteriorated results at three- and 15-month follow-ups, compared to the baseline results

[c]No changes: number of participants with no changes at three- and 15-month follow-ups, compared to the baseline results

[d]Improved: number of participants with improved results at three- and 15-month follow-ups, compared to the baseline results

**Table 8. Frequency distribution of SOLEC at baseline and follow-ups in WAD grade 2 and WAD grade 3.**

| Group | | | N[a] | Changes between baseline and three-month follow-up | | | Changes between baseline and 15-month follow-up | | |
|---|---|---|---|---|---|---|---|---|---|
| | | | | Deteriorated[b] | No changes[c] | Improved[d] | Deteriorated | No changes | Improved |
| WAD grade 2 | SOLEC, seconds at baseline | 0–10 s | 22 | 0 | 13 | 9 | 0 | 10 | 9 |
| | | 11–20 s | 8 | 6 | 0 | 2 | 3 | 4 | 1 |
| | | 21–28 s | 1 | 0 | 1 | 0 | 0 | 1 | 0 |
| | | 29–30 s | 6 | 1 | 5 | 0 | 1 | 4 | 0 |
| WAD grade 3 | SOLEC, seconds at baseline | 0–10 s | 16 | 0 | 12 | 4 | 0 | 8 | 6 |
| | | 11–20 s | 6 | 1 | 4 | 1 | 3 | 2 | 1 |
| | | 21–28 s | 4 | 1 | 1 | 2 | 1 | 0 | 2 |
| | | 29–30 s | 8 | 3 | 5 | 0 | 2 | 4 | 0 |

SOLEC at baseline is presented in four intervals: number of individuals reaching 0–10 seconds, 11–20 seconds, 21–28 seconds and 29–30 seconds

WAD: whiplash-associated disorder; WAD grade 2: whiplash-associated disorder, grade 2: neck pain and musculoskeletal findings during a physical examination; WAD grade 3: whiplash-associated disorder, grade 3: as grade 2, but with additional neurological findings

[a]N: number of participants in each interval at baseline

[b]Deteriorated: number of participants with deteriorated results at three- and 15-month follow-ups, compared to the baseline results

[c]No changes: number of participants with no changes at three- and 15-month follow-ups, compared to the baseline results

[d]Improved: number of participants with improved results at three- and 15-month follow-ups, compared to the baseline results

to treat. Even so, those with chronic WAD in the present study improved in terms of DHI score and dizziness during activity, and the NSE group also improved in dizziness at rest.

The prevalence of dizziness in the present study was 59% at baseline, which is in line with the 60% previously reported in a chronic WAD cohort by Treleaven et al. [7]. However, the percentages may vary between studies due to different populations and questions asked [49]. At three- and 15-month follow-ups, 81% and 70% respectively of those with dizziness at baseline had persistent problems of dizziness/unsteadiness despite the intervention. The results imply that other interventions are required, for example a specialised dizziness/balance exercise programme in addition to the neck-specific exercises.

Individuals reporting dizziness at baseline had higher NDI and more of them had WAD grade 3 compared to those without dizziness, showing that physiotherapists may need to pay extra attention to those with more severe WAD grades.

## Strengths and limitations

This study is unique and important, as no one has previously investigated the effects on dizziness/unsteadiness/balance in those with chronic WAD after using an internet exercise programme. Rehabilitation programmes for those with chronic moderate/severe WAD have seldom been investigated, and those individuals with WAD grade 3 are mostly excluded from studies investigating rehabilitation programmes.

One advantage of the study is that it is a prospective randomised controlled multicentre study in outpatient care, which is the level of care that most people with WAD seek. This makes it easier to implement the knowledge in healthcare facilities and makes the results more generalisable. The trial was a multicentre study over a fairly large geographic area, involving ten county councils, leading to less control over the interventions delivered. However, the physiotherapists received one day of training and written instructions, and could contact the project leader if questions arose.

The sample size and power calculation of the present secondary analyses of the RCT [24, 25] were calculated before data collection started, and were based on the main outcome

measurement in the RCT, NDI [24]. This can be seen as a limitation of the present study, where non-significant changes may be due to fewer participants (under-powered) in the present subgroup of dizziness, although most non-significant results were far from significant.

It may also be a limitation that the present population did not undergo a vertigo examination at a specialist clinic to ascertain the cause of the dizziness/unsteadiness. Since only a subgroup of 82 individuals from the original 140 study participants had dizziness and were further analysed in the present study, the effect of randomisation in the RCT was broken and the power was lower, although there were no significant group differences at baseline in the present sub-group between those randomised to NSEIT or NSE.

DHI is considered the golden standard for investigating self-reported dizziness, although its validity has been questioned [31] and Koppelaar-van Eijsden et al. [31] suggested that DHI should not be used as the only instrument in outcome studies of dizziness. In the present study, DHI was supplemented with measures of dizziness at rest and during activity, unsteadiness and physical measures of balance (SOLEC), which may be seen as a strength of the study.

## Conclusions

There were no significant group differences between NSEIT and NSE. Neck-specific exercise can decrease DHI score and dizziness during activity in chronic WAD (medium to large effect), and an internet programme can be used to reduce the number of visits to a physiotherapy clinic. The results of the study show that 59% of those with chronic WAD have dizziness/unsteadiness, and that those with more severe WAD (grade 3) more frequently report dizziness/unsteadiness, but also improved to a greater extent than those with WAD grade 2. Despite improvements, many participants still reported dizziness at 15-month follow-up, and additional balance training and/or vestibular exercise may be investigated for potential additional effect.

## Acknowledgments

We would like to thank all participants with whiplash-associated disorder, physiotherapists in primary care, the test leader and the research staff. The funder had no role in the study design, data collection and analysis, the decision to publish or the preparation of the manuscript.

## Author Contributions

**Conceptualization:** Anneli Peolsson, Gunnel Peterson.

**Data curation:** Gunnel Peterson.

**Formal analysis:** Gunnel Peterson.

**Funding acquisition:** Anneli Peolsson, Gunnel Peterson.

**Investigation:** Anneli Peolsson, Gunnel Peterson.

**Methodology:** Anneli Peolsson, Gunnel Peterson.

**Project administration:** Anneli Peolsson, Gunnel Peterson.

**Resources:** Anneli Peolsson, Gunnel Peterson.

**Software:** Anneli Peolsson.

**Visualization:** Gunnel Peterson.

**Writing – original draft:** Anneli Peolsson, Gunnel Peterson.

**Writing – review & editing:** Anneli Peolsson, Sara Wirqvist, Ann-Sofi Kammerlind, Gunnel Peterson.

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
