## [Decision Letter · Decision Letter 0]

22 May 2024

PONE-D-24-12568Effectiveness of neck-specific exercises with and without internet-based support on dizziness/unsteadiness in chronic whiplash-associated disorders: Secondary analyses of a randomised controlled trialPLOS ONE

Dear Dr. Peolsson,

Thank you for submitting your manuscript to PLOS ONE. After careful consideration, we feel that it has merit but does not fully meet PLOS ONE’s publication criteria as it currently stands. Therefore, we invite you to submit a revised version of the manuscript that addresses the points raised during the review process.

Please address carefully he comments of reviewer 2.

We look forward to receiving your revised manuscript.

Kind regards,

Andrea Martinuzzi

Academic Editor

PLOS ONE

“The study was founded by Sweden’s Innovation Agency, the Swedish Research Council, the Centre for Clinical

Research Sörmland at Uppsala University, the Regional Research Council Mid Sweden, the Medical Research Council of Southeast

Sweden, the county council of Östergötland, Sweden, and Linköping University, Sweden. All fundings was searched in open competition and money received to the Deparment.”

4. In the online submission form, you indicated that [Coded data will be received open reasonable request and after ethical permission. The data contain information about health].

Reviewers' comments:

Reviewer's Responses to Questions

**Comments to the Author**

1. Is the manuscript technically sound, and do the data support the conclusions?

Reviewer #1: Yes

Reviewer #2: Yes

2. Has the statistical analysis been performed appropriately and rigorously? 

Reviewer #1: Yes

Reviewer #2: Yes

3. Have the authors made all data underlying the findings in their manuscript fully available?

Reviewer #1: No

Reviewer #2: Yes

4. Is the manuscript presented in an intelligible fashion and written in standard English?

Reviewer #1: Yes

Reviewer #2: Yes

5. Review Comments to the Author

Reviewer #1: In this secondary analysis of a RCT the authors found no significant group differences between NSEIT and NSE. The methodology is rigorous and both strength and limitations (i.e. power calculation not performed for secondary outcomes, hence caution with interpretation of findings) have been reported in a balanced way.

According to PLOS ONE policy (publish according to methodological rigor rather than significance) I endorse the publication of this manuscript.

Reviewer #2: As the statistical reviewer I will focus on methods and reporting. Overall, this is a well conducted and reported study so i only have a few points to raise.

Minor

1) I appreciate more methodological info is provided elsewhere but can the authors add a sentence or two on power calculations and the sample size, since it's central to this as well, and some clarity is needed as to how it was arrived at.

2) this is a multi-centre study, so the authors could use random effects in their modelling, patients nested within centres. the authors do mention mixed linear models, but it was not clear what the random (intercept only?) component(s) was (were).

6. PLOS authors have the option to publish the peer review history of their article (what does this mean?). If published, this will include your full peer review and any attached files.

Reviewer #1: No

Reviewer #2: No

---

## [Author Response · Author response to Decision Letter 0]

6 Aug 2024

Response to Reviewers'

Ref: Submission PONE-D-24-12568

Effectiveness of neck-specific exercises with and without internet-based support on dizziness/unsteadiness in chronic whiplash-associated disorders: Secondary analyses of a randomised controlled trial

PLOS ONE

Thank you for reviewing and commenting on our manuscript. We believe that the manuscript has greatly improved after revision.

All answers from the authors are written in Times 14, blue text. The text that is altered in the original manuscript is marked as track changes (Revised Manuscript with Track Changes') as suggested together with an unmarked copy (Manuscript).

Page and line numbers are according to the original submission.

Answer to the Editor Dear Dr. Peolsson, Thank you for submitting your manuscript to PLOS ONE. After careful consideration, we feel that it has merit but does not fully meet PLOS ONE’s publication criteria as it currently stands. Therefore, we invite you to submit a revised version of the manuscript that addresses the points raised during the review process. Please address carefully he comments of reviewer 2.

ANSWER: Thank you for letting us revise the manuscript. The comments of reviewer 2 has been carefully addressed.

ANSWER: Thank you. The grant numbers and the main applicant of each grant is now in the manuscript as suggested. I will also correct the text in the “Funding information section” in the submission system.

“The study was founded by Sweden’s Innovation Agency, the Swedish Research Council, the Centre for Clinical

Research Sörmland at Uppsala University, the Regional Research Council Mid Sweden, the Medical Research Council of Southeast

Sweden, the county council of Östergötland, Sweden, and Linköping University, Sweden. All fundings was searched in open competition and money received to the Deparment.”

ANSWER:

Thank you. The following text was added at Line 528-530: The funders had no role in study design, data collection and analysis, decision to publish, or preparation of the manuscript. All fundings was searched in open competition and money received to the Department.

The Sweden’s Innovation Agency (Peterson, grant number 2018-02244)

The Swedish Research Council (Peolsson, grant number 2018-02476)

The Centre for Clinical Research Sörmland at Uppsala University, Sweden (Peterson, grant number DLL-36751, DLL-854531, DLL-930399, DLL-939813)

The Regional Research Council Mid Sweden (Peterson, grant number 838701)

The Medical Research Council of Southeast Sweden (Peolsson, grant number FORSS-939838, FORSS-563491; FORSS-650191)

The county council of Östergötland, Sweden, (Peolsson, grant number RÖ-724851, RÖ-602771)

Linköping University, Sweden (Peolsson, time for research included in the position as Professor)

4. In the online submission form, you indicated that [Coded data will be received open reasonable request and after ethical permission. The data contain information about health].

ANSWER: Our data cannot be made publicly available for both ethical and legal reasons as public availability would compromise patient privacy and information about their health. The statement is now included under the data availability section.

ANSWER:

Thank you. The information has been added at the end of the manuscript and citations has been made in the manuscript (Materials and Methods, page 6 and 7):

Supporting (S) information:

S1. Confirmation of ethical approval

S2. The ethical application in Swedish

S3. Published protocol, Peolsson et al. 2017

S4. Main outcome paper published Peterson & Peolsson 2023

S5. CONSORT 2010 checklist

S6. SPIRIT checklist WADIT dizziness 240229

S7. PLOS ONE Human Subjects Checklist 240301

6. Please review your reference list to ensure that it is complete and correct. If you have cited papers that have been retracted, please include the rationale for doing so in the manuscript text, or remove these references and replace them with relevant current references. Any changes to the reference list should be mentioned in the rebuttal letter that accompanies your revised manuscript. If you need to cite a retracted article, indicate the article’s retracted status in the References list and also include a citation and full reference for the retraction notice. ANSWER: The References has been proofread and are correct.

Reviewers' comments: Reviewer's Responses to Questions

Comments to the Author 1. Is the manuscript technically sound, and do the data support the conclusions? The manuscript must describe a technically sound piece of scientific research with data that supports the conclusions. Experiments must have been conducted rigorously, with appropriate controls, replication, and sample sizes. The conclusions must be drawn appropriately based on the data presented.

Reviewer #1: Yes

Reviewer #2: Yes

2. Has the statistical analysis been performed appropriately and rigorously?

Reviewer #1: Yes

Reviewer #2: Yes

3. Have the authors made all data underlying the findings in their manuscript fully available? The PLOS Data policy requires authors to make all data underlying the findings described in their manuscript fully available without restriction, with rare exception (please refer to the Data Availability Statement in the manuscript PDF file). The data should be provided as part of the manuscript or its supporting information, or deposited to a public repository. For example, in addition to summary statistics, the data points behind means, medians and variance measures should be available. If there are restrictions on publicly sharing data—e.g. participant privacy or use of data from a third party—those must be specified.

Reviewer #1: No Skriva in I manuskriptet under data tillgänglighet – det du skrev ovan.

Reviewer #2: Yes

4. Is the manuscript presented in an intelligible fashion and written in standard English? PLOS ONE does not copyedit accepted manuscripts, so the language in submitted articles must be clear, correct, and unambiguous. Any typographical or grammatical errors should be corrected at revision, so please note any specific errors here.

Reviewer #1: Yes

Reviewer #2: Yes

5. Review Comments to the Author Please use the space provided to explain your answers to the questions above. You may also include additional comments for the author, including concerns about dual publication, research ethics, or publication ethics. (Please upload your review as an attachment if it exceeds 20,000 characters)

Answer to Reviewer 1 and 2

Reviewer #1: In this secondary analysis of a RCT the authors found no significant group differences between NSEIT and NSE. The methodology is rigorous and both strength and limitations (i.e. power calculation not performed for secondary outcomes, hence caution with interpretation of findings) have been reported in a balanced way. According to PLOS ONE policy (publish according to methodological rigor rather than significance) I endorse the publication of this manuscript.

ANSWER: Thank you.

Reviewer #2: As the statistical reviewer I will focus on methods and reporting. Overall, this is a well conducted and reported study so i only have a few points to raise. ANSWER: Thank you.

Minor 1) I appreciate more methodological info is provided elsewhere but can the authors add a sentence or two on power calculations and the sample size, since it's central to this as well, and some clarity is needed as to how it was arrived at.

ANSWER:

Thank you. Statistical methods, page 9, Line 211: The text about power calculations in the RCT was extended as suggested: “Sample size and power calculations were carried out using Power Analysis and Sample Size (PASS, 13.0.8) software based on the primary outcome Neck Disability Index (NDI) in the RCT before data collection started (24, 25). To detect a between-group noninferiority margin of 7 (percentage units) with 1-sided α=.025 and β=.8, a total of 47 participants were needed in each group. To account for attrition, 70 participants were included in each group. The standardized between-group effect, that is, effect size at 3 and 15 months, was calculated as the mean differences between NSEIT and NSE divided by the pooled standard deviation (the weighted average of each group’s standard deviation). The standardized margin was calculated as the predefined noninferiority margin divided by the pooled standard deviation. NSEIT is considered noninferior to NSE in NDI if the lower limit (the 1-sided 95% CI of the effect size) does not exceed the standardized margin. Power calculation on secondary outcomes were not performed.”

2) this is a multi-centre study, so the authors could use random effects in their modelling, patients nested within centres. the authors do mention mixed linear models, but it was not clear what the random (intercept only?) component(s) was (were).

ANSWER: Thank you. Analyses of the continuous outcome variables (DHI, dizziness at rest and during activity, and unsteadiness) were undertaken using linear mixed models (LMMs), with time (baseline, three months, 15 months) and group (NSEIT, NSE) and group by time interaction (baseline, three months, 15 months) as fixed effects. We used LMM with maximum likelihood parameter estimation to avoid participant exclusion due to single missing data points, which according to the University statistician is a better alternative than using a complete case analysis as repeated measures ANOVA. We were interested in the differences between the randomisation groups (NSE, NSEIT) and differences over time (from baseline to follow-ups). Random effects were not included as we were not interested (not the aim of the study) in geographical/ centres differences (57 treating physiotherapists in 10 different counties/ regions in the south and middle part of Sweden). The consequence of cluster analysis would be too small groups in the present study group. We added additional information in the manuscript, page 8, line 190 ” Analyses of the continuous outcome variables (DHI, dizziness at rest and during activity, and unsteadiness) were undertaken using linear mixed models (LMMs), with time (baseline, three months, 15 months) and group (NSEIT, NSE) and group by time (baseline, three months, 15 months) as fixed effects. Restricted maximum likelihood estimate was used in the LMMs, allowing all participants with at least one observation to be included in the models, under the assumption of data missing at random.” to make it more clear for the reader.

6. PLOS authors have the option to publish the peer review history of their article (what does this mean?). If published, this will include your full peer review and any attached files. Do you want your identity to be public for this peer review? For information about this choice, including consent withdrawal, please see our Privacy Policy.

Reviewer #1: No

Reviewer #2: No

ANSWER: Figure file (Figure 1. Consort flow-chart) has been uploaded to the Preflight Analysis and Conversion Engine (PACE) digital diagnostic tool.

---

## [Decision Letter · Decision Letter 1]

15 Sep 2024

Effectiveness of neck-specific exercises with and without internet-based support on dizziness/unsteadiness in chronic whiplash-associated disorders: Secondary analyses of a randomised controlled trial

PONE-D-24-12568R1

Dear Dr. Peolsson,

We’re pleased to inform you that your manuscript has been judged scientifically suitable for publication and will be formally accepted for publication once it meets all outstanding technical requirements.

Kind regards,

Andrea Martinuzzi

Academic Editor

PLOS ONE

Additional Editor Comments (optional):

Reviewers' comments:

Reviewer's Responses to Questions

**Comments to the Author**

1. If the authors have adequately addressed your comments raised in a previous round of review and you feel that this manuscript is now acceptable for publication, you may indicate that here to bypass the “Comments to the Author” section, enter your conflict of interest statement in the “Confidential to Editor” section, and submit your "Accept" recommendation.

Reviewer #2: All comments have been addressed

2. Is the manuscript technically sound, and do the data support the conclusions?

Reviewer #2: Yes

3. Has the statistical analysis been performed appropriately and rigorously? 

Reviewer #2: Yes

4. Have the authors made all data underlying the findings in their manuscript fully available?

Reviewer #2: Yes

5. Is the manuscript presented in an intelligible fashion and written in standard English?

Reviewer #2: Yes

6. Review Comments to the Author

Reviewer #2: I am satisfied with the authors' responses and the resulting changes to the paper, following the first stage of peer review.

7. PLOS authors have the option to publish the peer review history of their article (what does this mean?). If published, this will include your full peer review and any attached files.

Reviewer #2: No

---

## [Editor Report · Acceptance letter]

25 Sep 2024

PONE-D-24-12568R1 

PLOS ONE

Dear Dr. Peolsson, 

I'm pleased to inform you that your manuscript has been deemed suitable for publication in PLOS ONE. Congratulations! Your manuscript is now being handed over to our production team.

Kind regards, 

on behalf of

Dr. Andrea Martinuzzi 

Academic Editor

PLOS ONE